# Enhance the Transferability of Adversarial Attacks through Channel Pruning

## Abstract

Recent studies have shown that neural networks are vulnerable to adversarial attacks, where attackers generate adversarial samples by imposing tiny noise. The tiny noise can not misguide human perception, though leading the neural networks to generate wrong predictions. Transfer-based black-box attacks play a more significant role in recent studies due to their more realistic setting and considerable progress in performance. Previous studies have shown that some different channels of the same layer in convolution neural networks (CNN) contain lots of repetitive information, and we find that existing transferable attacks tend to exploit those redundant features more, which limits their transferability. Hence, we advocate using channel pruning and knowledge distillation to conduct model augmentation. In addition, we introduce a method of regularization on the gradients of intermediate feature maps of augmented models, which further enhances the transferability of our method. Comprehensive experiments demonstrate that imposing our method of model augmentation on existing methods can significantly improve the transferability of adversarial attacks in untargeted or targeted scenarios. Furthermore, our method outperforms state-of-the-art model augmentation techniques without the usage of additional training datasets.

## 1 Introduction

Deep Neural Networks (DNNs) have become the mainstream backbone for various computer vision tasks, including image classification (Krizhevsky et al., 2012), object detection (Girshick et al., 2015), segmentation (Farabet et al., 2012), and so on. However, prior studies (Goodfellow et al., 2014) have shown that DNNs are vulnerable to adversarial attacks, where tiny noise imperceptible to humans is imposed on original images to misguide the DNN model prediction. Therefore, research regarding adversarial attacks is not only crucial for the security and robustness concern of DNN models (Carlini & Wagner, 2017) but also helpful for distinguishing between humans and computers (Shi et al., 2021; Lee et al.).

Adversarial attacks are categorized into two different settings. In the white-box setting, attackers can fully access information about the target model, including model architecture, parameters, training data, and loss function. On the other hand, Recent studies have mainly focused on the black-box setting, which is more realistic in the real world. It requires that attackers do not have any internal details about target models. In most situations, we need to generate adversarial samples on a surrogate model and transfer them to other target models, called transfer-based attacks. Lots of studies focus on improving the transferability of adversarial attacks in the black-box setting.

To boost the transferability of adversarial attacks, we design a method of model augmentation inspired by the internal characteristic of most Convolutional Neural Networks (CNN) - based models. Prior studies (Roheda & Krim, 2020) have shown that CNN models are often "over-parameterized," which means that they contain lots of redundant channels in each layer. Feature maps given by some channels are highly close to those given by other channels or a composite of other channels. From the top row of Figure 1, we can clearly observe that some of the shallow feature maps from the original model contain rich information but look very similar to each other. On the contrary, some of them are vague and seem to only contain weak features about the object in the input image. If we only conduct adversarial attacks based on original CNN models, the adversarial samples tend to "overfit" on those highly repetitive features. Namely, adversarial samples disrupting those features

more will achieve locally optimized results on surrogate models but might not cause the same effect on target models. Moreover, via the illustration from the bottom row of Figure 1, we find that the attacks on original models disrupt the feature of a specific channel with much more intensity than they do on the other ones. Namely, the attacks on original models are not distributed evenly on each channel. This behavior makes the attacks less robust and prone to model-specific features.

To address these problems, we propose Gradient Regularization and Structured Channel Pruning (GRASP). We apply channel pruning to a surrogate model and then ensemble multiple copies of the pruned networks with different pruning rates to form an enhanced version of the source model. (Section 4.1) We reference previous studies (He et al., 2019) on channel pruning and adopt the previous criterion proposed to select the "unnecessary" channels, which will be pruned afterward.

Most of the channel pruning methods need a huge training dataset to fine-tune the pruned models. Otherwise, the pruned models usually perform too badly to be applicable. This requirement is not practical in real scenarios because attackers usually have no access to the training dataset of target models. Hence, we utilize the technique of few-shot knowledge distillation (Xu et al., 2020) instead of traditional fine-tuning to recover the classification accuracy of the pruned models (Section 4.2). To balance the importance of each channel in adversarial attacks, we further introduce a new method of regularization to the gradients of intermediate feature maps (Section 4.3). Through the visualization in Figure 1, we find that applying our methods of initial parameter optimization and regularization makes the pruned surrogate models contain denser information. Namely, those vague and weak features contain richer information related to the geometry of objects in input images. More importantly, the distortion of the feature maps caused by adversarial attacks is more evenly distributed to all channels except for the repetitive ones, which are pruned beforehand.

In summary, our primary contributions include:

- We propose a novel approach of model augmentation leveraging channel pruning, knowledge distillation, and gradient regularization to improve the transferability of adversarial attacks.
- We provide an intuitive perspective and explanation of the limits of the existing algorithm, which is related to the "distribution" of attacks on different channels. In addition, we introduce a method to overcome the limits.
- Comprehensive experiments have shown that applying our approach to existing algorithms of adversarial attacks significantly improves transferability. Moreover, our method outperforms the state-of-the-art method of model augmentation without additional training data.

## 2 RELATED WORK

### 2.1 ADVERSARIAL TRANSFERABILITY

Current progress in the transferability of adversarial attacks mainly relies on input augmentation-based approaches, gradient-based approaches and model enhancement. In input augmentation-based methods (Xie et al., 2019; Dong et al., 2019; Lin et al., 2019; Wu et al., 2021), we apply some transformation to the original inputs to make them contain more versatile and diversified information, then using an ensemble of transformed images for optimization.

On the other hand, gradient-based methods (Dong et al., 2018; Lin et al., 2019; Wang & He, 2021; Qin et al., 2022; Xiong et al., 2022; Xu et al., 2023) focus on modifying and regularizing the process of gradient propagation to improve its stability and avoid poor local optima. For example, Wang & He (2021) consider the variance of gradients in the previous iteration to stabilize the direction of gradients in each iteration. Qin et al. (2022) propose a bi-level min-max optimization process. Xu et al. (2023) adjust the path of backpropagation by decomposing convolution kernels and structural reparameterization.

We mainly focus on model augmentation technique (Li et al., 2020; Liang & Xiao, 2023; Li et al., 2023; Wu et al., 2024; Wang et al., 2024), where some modification is imposed on the original source model to make the adversarial attacks based on it more transferable. Usually, an ensemble of multiple modified versions of the source model is used. Note that the model augmentation technique can be combined with other input-based or gradient-based approaches with great flexibility. For

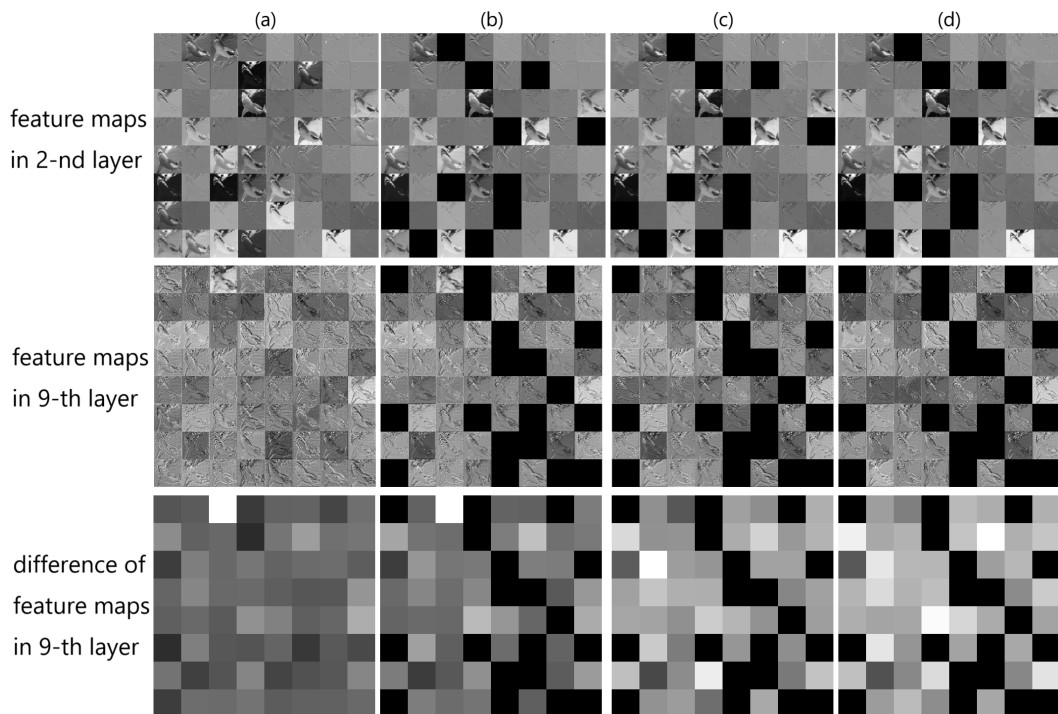

Figure 1: The top row shows the feature maps from the 2-nd layer of ResNet-50. The middle row shows the feature maps from the 9-th layer of ResNet-50. The bottom row illustrates the distance between the feature maps before attacks and after attacks. (Deeper color represents smaller distance.) In all the figures, each grid represents a channel. (a) the original model (b) the pruned model (c) the pruned model with initial parameter optimization (d) the pruned model with initial parameter optimization and regularization of intermediate feature maps.
Overall, the color is brighter and more balanced in the (c) and (d) column of the bottom row, which presents that applying our method makes the distortion of feature in each channel more even.

example, Liang & Xiao (2023) advocate using an ensemble of different stylized surrogate models by inserting an adaptive instance normalization layer into vanilla models. Li et al. (2023) enhance the diversity in surrogate models by training and attacking a Bayesian model. Wu et al. (2024) leverage Lipschitz regularization on the loss landscape of surrogate models to make the optimization process smoother and more controllable. Wang et al. (2024) also leverage some pruning of model weights or downsizing technique to generate model augmentation. However, they propose to prune the surrogate models with an relatively unstructural way similar to Dropout (Srivastava et al., 2014), where a single weight is pruned at a time. We advocate prune the surrogate models with a more structural way, where a whole channel is pruned at a time. Our method might be more aligned to the problem of over-concentration on specific channels in transfer attacks.

## 2.2 CHANNEL PRUNING

Channel pruning is a dominant approach to compress CNN-based neural networks. The simplest way to accomplish this is to eliminate some channels in each layer using certain criteria based on the parameters / feature maps and sparsity constraints. For example, Zhuang et al. (2018) keep the most discriminative channels in each layer. He et al. (2019) filter the channels whose parameters are closer to the geometric median of all the channels in the same layer. They claim that those channels are more likely to be redundant and less important because they are easier to be composites of other channels. Sui et al. (2021) eliminate the channels with less independence, which is quantified by a novel inter-channel metric. Moreover, there are also some studies that advocate a more dynamic way to conduct channel pruning. For example, Li et al. (2022) incorporate the ideas of neural architecture search (NAS) and search out a new family of light-weight networks automatically by

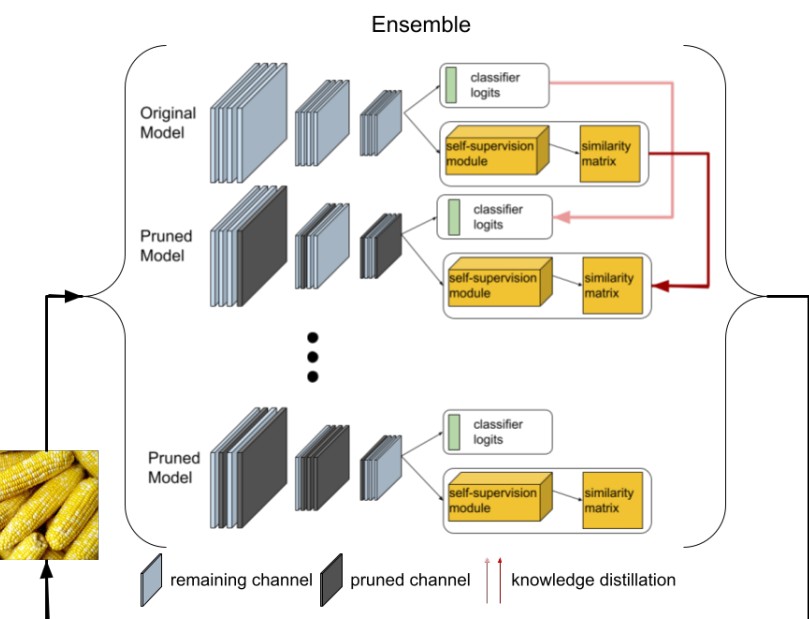

Figure 2: The overview of our method. We leverage channel pruning and initial parameter optimization to generate multiple augmented models, updating adversarial samples using an ensemble of them.

adding some additional layers and reparameterization. Hou et al. (2022) dynamically and repeatedly prune and reallocate the channels in each epoch throughout the training process. For simplicity and efficiency, we adopt the method proposed by He et al. (2019) to conduct channel pruning. Besides, the thoughts behind this work is quite correspondent to our intuition, where the redundant channels are more likely to hinder the transferability of adversarial attacks.

## 3 PRELIMINARIES

**Attack objective.** Given a surrogate model $f_\theta$ and a benign image $x$ with a ground truth label $y$. The objective of adversarial attack is to find an adversarial sample that maximizes the loss function and meanwhile resembles the benign image $x$. The problem can be formulated as:

$$\max_\delta L(f_\theta(x + \delta), y) \quad s.t. \quad ||\delta|| \leq \epsilon$$

where $L$ denotes the loss function and $\epsilon$ denotes the budget of adversarial attack. In this formulation, $x + \delta$ is the adversarial sample we craft.

We define the transferability of adversarial attack as the rate of the adversarial samples which successfully misguide the prediction of a target model. In the black-box setting, we can not know the architecture and parameters of the target model when we construct adversarial samples.

**Iterative Attack.** Iterative fast gradient sign method (I-FGSM) Kurakin et al. (2018) is a popular method to solve this problem. In each iteration, the adversarial sample is updated according to the gradient of the loss function with respect to the current sample. This method can be formulated as:

$$x_{adv}^{t+1} = x_{adv}^t + \alpha \cdot sign(\nabla_x L(f_\theta(x_{adv}^t), y)$$

where $t$ is the index of iteration and $\alpha$ is the learning rate. To ensure that the attack budget is not exceeded, a clip function is used in each iteration.

**Model Augmentation.** Model augmentation refers to a technique of using an ensemble of $M(M > 1)$ different models, each of which is generated from the same pre-trained model and architecture

but slightly different from each other. The ensemble of these augmented models is used to generate adversarial examples. To be specific, the equation of the iterative attack is modified to

$$x_{adv}^{t+1} = x_{adv}^t + \alpha \cdot sign(\nabla_x L(\frac{1}{M} \sum_{i=1}^{M} f_i(x_{adv}^t), y)$$

where $f_i$ denotes the output of the $i$-th augmented model.

**Contrastive Learning.** Contrastive learning is a common pretext task used in self-supervised learning. (Chen et al., 2020) In this task, we apply some augmentation to all the training images and train the neural network to match the augmented images and the original images within the same batch. We can add self-supervision (SS) modules to perform the task on the original networks. Given a mini-batch of data containing $B$ inputs $\{x_i\}_{i=1,2,...,B}$. We apply transformation to each of the $x_i$, obtaining the augmented inputs $\{x_i'\}_{i=1,2,...,B}$. The SS module maps each of the $x_i$ and $x_i'$ to feature vectors $z_i$ and $z_i'$. Then, we can organize the cosine similarity between each possible pair of $(z_i, z_j')$ into a $B \times B$ similarity matrix $M$. In other words, we have

$$M_{i,j} = \text{cosine\_similarity}(z_i, z_j'),$$

which is the target for optimization in contrastive learning.

## 4 METHODOLOGY

In this section, we describe the detailed process to generate an augmented model. Note that the process will be repeated for $M$ times if we want to generate $M$ augmented models. First, we need to perform channel pruning to eliminate redundant channels. (described in Section 4.1) Then, we need to perform initial parameter optimization to make the pruned model more powerful. (described in Section 4.2). Finally, we introduce a method of regularization on the gradients to intermediate feature maps, which is performed during initial parameter optimization. (described in Section 4.3). We show the big picture of our method in Figure 2.

### 4.1 CHANNEL PRUNING

To avoid overfitting on redundant channels or model-specific features of the source CNN model, we introduce the technique of channel pruning to perform model augmentation. For each augmented model, we randomly sample different pruning rates, namely the rate of channels to be pruned, in each layer based on the predetermined lower bound ($r_{min}$) and upper bound ($r_{max}$).

After we sample a pruning rate for each layer in an augmented model, we select the corresponding number of channels and eliminate them in each layer. Based on the prior channel running method He et al. (2019), we select the channels with minimum summation of the distance with other channels. We denote $F_{i,j}$ as the output feature map in the $j$-th channel of the $i$-th layer. The criterion for the selection of pruned channels can be formulated as:

$$j^* = \arg\min_j \sum_{j' \in [1,n_i]} ||F_{i,j} - F_{i,j'}||_2 \quad s.t. \quad j \in [1, n_i]$$

where $n_i$ denotes the number of channels in the $i$-th layer.

### 4.2 INITIAL PARAMETER OPTIMIZATION

In classical methods of channel pruning, fine-tuning of the model on training datasets is required to maintain its performance. However, since the training dataset for target models might not be available, fine-tuning on the original training dataset for each augmented model is not realistic in practice. Hence, we incorporate few-shot knowledge distillation techniques to fulfill the generation of augmented models without additional data. Namely, we only perform knowledge distillation on the target dataset (i.e. the dataset consisting of 2000 images we are going to attack) to generate augmented models.

To train the pruned models with a small dataset while achieving an acceptable performance on image classification, we leverage knowledge distillation in order to transfer the knowledge from original

models to pruned models. Prior work (Hinton et al., 2015) has shown that classification logits predicted by a well-pretrained model can provide additional information. Making a student model (pruned model in our case) mimic its logits can let the student model acquire additional information than that provided by ground truth labels. Hence, to train the pruned models, we use KL-divergence between the logits predicted by the original models and the pruned models in addition to cross-entropy loss (CE loss), which is typically used in learning classification tasks.

In addition to conventional knowledge distillation, we incorporate the technique proposed by Xu et al. (2020), which employs contrastive prediction Chen et al. (2020) as an auxiliary task alongside the original classification task to extract more comprehensive information from pre-trained models. To implement the auxiliary task, we attach a lightweight self-supervision (SS) module to both the original and pruned model backbones. Initially, we train the SS module of the original model using both normal and transformed inputs. For contrastive learning, we apply three types of transformations: rotation, randomly masking individual pixels, and a simple adversarial attack generated using FGSM (Goodfellow et al., 2014).

We train the pruned model using a combination of four losses, which are the cross entropy loss for normal inputs ($L_{ce}$), the knowledge distillation loss ($L_{kd}$), the KL-divergence loss between the SS module's outputs ($L_{ss}$), and the knowledge distillation loss for transformed inputs ($L_T$). We denote the classifier outputs of the original model and the pruned model as $v^o(x)$ and $v^p(x)$ respectively, and denote the similarity matrix post-processed with softmax for the original model and the pruned model as $A^o$ and $A^p$, respectively (note that $x$ denotes the input data). $v^o(x)$ and $v^p(x)$ are vectors with $C$ dimensions, where $C$ is the number of classes. $A^o$ and $A^p$ are $B \times B$ matrices, where $B$ is the batch size. $L_{kd}$, $L_{ss}$ and $L_T$ can be formulated as:

$$L_{kd} = -\sum_{x \in D} \sum_{i=1}^{C} v_i^o(x) log(v_i^p(x))$$

$$L_{ss} = -\sum_{i,j} A_{i,j}^o log(A_{i,j}^p)$$

$$L_T = -\sum_{x \in T(D)} \sum_{i=1}^{C} v_i^o(x) log(v_i^p(x))$$

where D is the dataset of the normal images and T(D) is the dataset of all the transformed images.

### 4.3 REGULARIZATION OF INTERMEDIATE FEATURE MAPS

To further reduce the variance of performance between different target models, we introduce a technique of regularization. Wang et al. (2021) suggest that the gradients of logits output with respect to a specific intermediate feature map of source models are related to the "importance" of the feature map in adversarial attacks. The feature maps with larger values of gradients tend to play a more critical role when it comes to adversarial attacks transferred to various target models. That is, disrupting those feature maps can lead to stronger transferability. In addition, Zhang et al. (2023) have shown that regularizing the back-propagated gradients of internal blocks by eliminating extreme value can improve the transferability for both convolution-based and transformer-based surrogate/target models.

Hence, we regularize the back-propagated gradients in a layer of intermediate feature maps from source models during initial parameter optimization. If we fine-tune the model to shrink the variance of the gradients from different channels, the "importance" of each channel in that layer is more balanced. In addition, model-specific features, which are harmful to the transferability of adversarial attacks, can be modified to more generic ones through the regularization process. We realize the regularization process by using an additional loss function $L_{reg}$, formulated as:

$$L_{reg} = \sum_{x \in D} var(\nabla_{\boldsymbol{F}_l^p} v^p(x))$$

where $F_l^p$ denotes the parameters of the $l$-th layer in the pruned model. The overall loss function becomes:

$$L = w_{ce}L_{ce} + w_{kd}L_{kd} + w_{ss}L_{ss} + w_T L_T + w_{reg}L_{reg}$$

where $w_{ce}, w_{kd}, w_{ss}, w_T, w_{reg}$ are hyper-parameters.

## 5 EXPERIMENTS

### 5.1 EXPERIMENTAL SETTINGS

**Dataset and Evaluated Models.** We conduct our experiments on the ImageNet (Russakovsky et al., 2015)-compatible dataset including 2000 images. We randomly sample 2 images from each category of the ImageNet dataset. The images sampled are almost correctly classified by all the evaluated models. For untargeted attacks, we use ResNet-50, ResNet-101 (He et al., 2016), Inception-v3 (Szegedy et al., 2016) as the surrogate models. For targeted attacks, we use ResNet-50, VGG-16 (Simonyan & Zisserman, 2014), DenseNet-121 (Huang et al., 2017), and Inception-v3 as the surrogate models. In addition, we use more diverse architectures for the target models, including some transformer-based models such as ViT-S and ViT-B (Dosovitskiy, 2020).

**Baseline Methods.** We conduct our experiments based on some commonly used attacking methods by comparing their results with or without injecting our techniques. The baseline methods include MI-FGSM (Dong et al., 2018), DI-FGSM (Xie et al., 2019) , SINI-FGSM (Lin et al., 2019) and VNI-FGSM (Wang & He, 2021). In addition, we also compare our methods to state-of-the-art (SOTA) methods of model augmentation in adversarial attacks.

**Implementation Details.** We adopt cross-entropy (CE) loss for adversarial attacks in all the experiments. Following the prior works (Wang et al., 2024; Liang & Xiao, 2023; Wang & He, 2021), we set the budget for adversarial perturbation $\epsilon$ as $16/255$. The step size is set as $1.6/255$. The number of steps is set as 200 in DI-FGSM, while it is set as 10 in the other baseline methods. The momentum decay in MI-FGSM is 1, and the number of scale copies in SINI-FGSM is 5. The probability of applying input diversity is set as 0.7 in DI-FGSM. The number of augmented models $M$ is set as 3. For all experiments, we use random mask as transformation in the initial parameter optimization. The training of augmented models (initial parameter optimization) uses the SGD optimizer. We train the SS module of the original models with 100 epochs. In addition, we train the pruned models with 130 epochs without regularization of intermediate feature maps or with 200 epochs with regularization.

### 5.2 THE EVALUATION OF UNTARGETED ATTACKS

We evaluate the performance of our method on various baseline methods in untargeted attacks. The results are shown in table 1. "+GRASP w/o Reg" indicates the application of our method without regularizing intermediate feature maps,, while "+GRASP" includes regularization. It is evident that our method significantly enhances the effectiveness of several existing methods, and regularizing intermediate feature maps further improves performance. For example, in transfer attacks from ResNet-50 to other CNN models (ResNet-101, Inception-v3, Inception-v4), applying our method with regularization increases the average attack success rate by 31.9% (MI), 14.1% (SINI), 4.5% (VNI) and 10.0% (DI), respectively.

Although transfer attacks between CNN-based and transformer-based architectures are generally more challenging, our method with regularization also boosts performance in ResNet-50 $\Rightarrow$ ViT-S and ViT-B by 28.4% (MI), 22.4% (SINI), 16.0% (VNI) and 31.5% (DI), respectively. Similar results are observed when using ResNet-101 as the surrogate model. Even though Inception-v3 often shows lower transferability, our method still significantly enhances the baseline attacks. In nearly all cases, our method with regularization outperforms the unregularized version, confirming that reducing the variance of gradients in intermediate feature maps is crucial for improving the transferability of adversarial attacks.

### 5.3 THE EVALUATION OF TARGETED ATTACKS

Similar to the above experiments, we evaluated the performance of our method on various baseline methods in targeted attacks. Following the previous experiments, we evaluate the performance of our method on various baseline methods in targeted attacks, with results shown in Table 2. Applying our model augmentation method significantly improves transferability in nearly all cases. Combining our approach with commonly used input augmentation methods yields remarkably strong performance in targeted attacks. For instance, by applying our method with regularization, the attack success rate of DI increases 59.3% (ResNet-50 $\Rightarrow$ VGG-16), 67.6% (ResNet-50 $\Rightarrow$ DenseNet-121), 49.4% (ResNet-50 $\Rightarrow$ Inception-v3) and 32.2% (Densenet-121 $\Rightarrow$ ResNet-50), respectively. This

Table 1: Untargeted attack success rate (%) of the baseline methods with our technique.

| Attack | ResNet-50 $\Rightarrow$ | | | | |
|---|---|---|---|---|---|
| | ResNet-101 | Inception-v3 | Inception-v4 | ViT-S | ViT-B |
| MI/ +GRASP w/o Reg/ +GRASP | 60.3/84.2/**88.8** | 56.2/84.4/**89.2** | 50.4/80.0/**84.6** | 28.5/56.8/**61.0** | 19.2/39.4/**43.4** |
| SINI/ +GRASP w/o Reg/ +GRASP | 82.4/94.3/**95.1** | 82.7/94.9/**95.6** | 77.9/93.3/**94.6** | 41.9/65.8/**67.3** | 26.1/42.5/**45.5** |
| VNI/ +GRASP w/o Reg/ +GRASP | 84.0/84.8/**87.0** | 82.7/84.4/**86.5** | 78.8/84.0/**85.4** | 48.8/66.0/**66.8** | 33.9/**51.3**/47.9 |
| DI/ +GRASP w/o Reg/ +GRASP | 89.5/97.4/**97.6** | 88.0/98.2/**98.6** | 85.3/**97.5**/96.7 | 47.7/79.0/**82.5** | 34.3/55.0/**62.4** |
| Attack | ResNet-101 $\Rightarrow$ | | | | |
| | ResNet-50 | Inception-v3 | Inception-v4 | ViT-S | ViT-B |
| MI/ +GRASP w/o Reg/ +GRASP | 69.6/92.3/**93.7** | 55.8/86.4/**87.9** | 49.8/79.3/**82.3** | 29.6/60.5/**63.7** | 19.4/42.9/**47.7** |
| SINI/ +GRASP w/o Reg/ +GRASP | 86.9/96.3/**96.5** | 79.5/95.0/**95.1** | 74.3/92.0/**92.7** | 45.4/67.2/**69.7** | 29.2/45.1/**47.2** |
| VNI/ +GRASP w/o Reg/ +GRASP | **88.1**/85.1/86.1 | 80.8/81.1/**83.9** | 76.0/79.8/**81.5** | 52.6/64.5/**65.4** | 35.0/49.5/**50.1** |
| DI/ +GRASP w/o Reg/ +GRASP | 92.1/98.9/**99.3** | 83.8/97.8/**98.3** | 81.5/95.7/**96.7** | 47.5/77.6/**81.0** | 31.8/55.4/**61.4** |
| Attack | Inception-v3 $\Rightarrow$ | | | | |
| | ResNet-50 | ResNet-101 | Inception-v4 | ViT-S | ViT-B |
| MI/ +GRASP w/o Reg/ +GRASP | 37.1/58.5/**63.9** | 31.0/50.7/**56.9** | 47.1/71.9/**74.3** | 15.5/30.2/**38.0** | 10.0/18.9/**26.4** |
| SINI/ +GRASP w/o Reg/ +GRASP | 61.0/77.9/**79.6** | 53.4/71.4/**73.7** | 74.8/**90.1**/89.8 | 28.5/43.1/**49.6** | 18.6/29.4/**33.1** |
| VNI/ +GRASP w/o Reg/ +GRASP | 52.3/71.9/**74.6** | 50.7/67.2/**70.4** | 76.5/**85.1**/84.8 | 30.9/46.2/**51.3** | 20.6/32.4/**38.2** |
| DI/ +GRASP w/o Reg/ +GRASP | 55.0/75.5/**81.0** | 45.9/67.1/**73.0** | 71.6/**91.7**/90.1 | 19.9/37.0/**44.1** | 12.0/23.5/**29.7** |

Table 2: Targeted attack success rate (%) of the baseline methods with our technique.

| Attack | ResNet-50 $\Rightarrow$ | | | | |
|---|---|---|---|---|---|
| | VGG-16 | DenseNet-121 | Inception-v3 | ViT-S | ViT-B |
| MI/ +GRASP w/o Reg/ +GRASP | 1.5/12.4/**13.3** | 2.2/21.0/**26.3** | 0.7/**6.0**/4.9 | 0.1/**1.4**/0.6 | 0.1/0.4/**0.6** |
| SINI/ +GRASP w/o Reg/ +GRASP | 6.8/21.4/**22.5** | 13.1/38.0/**38.6** | 3.7/13.3/**14.4** | 0.6/0.3/**2.1** | 0.4/0.9/**1.2** |
| VNI/ +GRASP w/o Reg/ +GRASP | 4.5/**24.6**/22.0 | 8.3/**30.5**/26.3 | 3.0/**14.6**/13.1 | 0.4/**5.2**/4.4 | 0.6/**3.1**/2.2 |
| DI/ +GRASP w/o Reg/ +GRASP | 12.6/71.2/**71.9** | 20.3/87.2/**87.9** | 5.3/48.7/**54.7** | 0.5/7.0/**8.2** | 0.1/**6.1**/5.6 |
| Attack | VGG-16 $\Rightarrow$ | | | | |
| | ResNet-50 | DenseNet-121 | Inception-v3 | ViT-S | ViT-B |
| MI/ +GRASP w/o Reg/ +GRASP | 0.3/3.9/**5.3** | 0.3/3.5/**4.3** | 0.3/2.9/**3.5** | 0.0/**0.4/0.4** | 0.1/0.1/0.1 |
| SINI/ +GRASP w/o Reg/ +GRASP | 2.4/8.3/**8.8** | 2.2/8.6/**10.6** | 1.4/6.4/**7.3** | 0.1/**0.8**/0.5 | 0.2/**0.5/0.5** |
| VNI/ +GRASP w/o Reg/ +GRASP | 3.3/**10.6**/9.1 | 2.2/11.8/**13.3** | 1.9/**9.6**/9.2 | 0.6/**2.1**/1.7 | 0.1/**1.3**/1.1 |
| DI/ +GRASP w/o Reg/ +GRASP | 0.8/**15.9**/14.1 | 0.5/**14.6**/14.1 | 0.3/**11.0**/10.9 | 0.0/**1.1**/0.3 | 0.0/0.4/**0.5** |
| Attack | DenseNet-121 $\Rightarrow$ | | | | |
| | ResNet-50 | VGG-16 | Inception-v3 | ViT-S | ViT-B |
| MI/ +GRASP w/o Reg/ +GRASP | 2.0/**7.8**/6.5 | 0.8/3.6/**3.8** | 0.3/2.2/**2.6** | 0.0/0.5/**0.6** | 0.0/**0.4**/0.3 |
| SINI/ +GRASP w/o Reg/ +GRASP | 7.8/15.3/**16.2** | 3.7/6.7/**7.3** | 3.0/6.3/**7.0** | 0.2/1.1/**1.7** | 0.4/0.8/**0.9** |
| VNI/ +GRASP w/o Reg/ +GRASP | 4.2/11.6/**14.1** | 2.3/7.5/**7.9** | 1.9/5.3/**6.3** | 0.4/1.7/**2.0** | 0.3/1.1/**1.4** |
| DI/ +GRASP w/o Reg/ +GRASP | 18.7/47.2/**50.9** | 4.5/23.2/**25.0** | 3.8/20.0/**22.8** | 0.3/4.1/**5.0** | 0.2/2.6/**4.4** |
| Attack | Inception-v3 $\Rightarrow$ | | | | |
| | ResNet-50 | DenseNet-121 | VGG-16 | ViT-S | ViT-B |
| MI/ +GRASP w/o Reg/ +GRASP | 0.4/1.4/**1.7** | 0.6/1.5/**2.0** | 0.5/1.6/**1.7** | 0.1/0.1/**0.4** | 0.0/**0.2**/0.0 |
| SINI/ +GRASP w/o Reg/ +GRASP | 1.4/**4.4**/4.1 | 1.5/**4.8**/4.6 | 1.6/**4.5**/3.8 | 0.1/**0.6**/0.5 | 0.1/**0.2/0.2** |
| VNI/ +GRASP w/o Reg/ +GRASP | 1.0/4.4/**5.1** | 1.0/**4.4**/4.0 | 1.2/4.5/**4.7** | 0.2/1.1/**1.2** | 0.3/**0.7/0.7** |
| DI/ +GRASP w/o Reg/ +GRASP | 1.3/12.3/**14.3** | 1.1/14.9/**16.1** | 2.2/**17.2**/16.1 | 0.0/0.7/**0.8** | 0.0/0.7/**0.8** |

demonstrates that strong targeted attacks are achievable even with surrogate models from a single architecture type. Although attacks using VGG-16 or Inception-v3 as the surrogate model or targeting transformer-based models typically exhibit weaker performance, our method still shows significant improvement in these cases.

## 5.4 COMPARISON WITH OTHER MODEL AUGMENTATION METHODS

For a fair comparison, we evaluate our method against SOTA model augmentation methods that do not require additional training datasets. The comparison includes StyLess (Liang & Xiao, 2023) and DWP (Wang et al., 2024). We use MI-FGSM as the baseline for untargeted attacks and DI-

Table 3: Comparison of the untargeted attack success rate (%) with different model augmentation technique

| Attack | ResNet-50 ⇒ | | | | |
|---|---|---|---|---|---|
| | ResNet-101 | Inception-v3 | Inception-v4 | ViT-S | ViT-B |
| MI +StyLess | 71.1 | 75.6 | 67.7 | 40.4 | 24.7 |
| MI +DWP | 73.9 | 43.0 | 38.8 | 22.2 | 10.7 |
| MI +GRASP | **88.8** | **89.2** | **84.6** | **61.0** | **43.4** |

Table 4: comparison of the targeted attack success rate (%) with different model augmentation technique

| Attack | ResNet-50 ⇒ | | | | |
|---|---|---|---|---|---|
| | VGG-16 | DenseNet-121 | Inception-v3 | ViT-S | ViT-B |
| DI +StyLess | 34.3 | 45.2 | 18.5 | 2.1 | 1.2 |
| DI +DWP | 65.6 | 75.0 | 34.1 | 5.8 | 2.8 |
| DI +GRASP | **71.9** | **87.9** | **54.7** | **8.2** | **5.6** |

FGSM for targeted attacks, running the authors' provided code with default configurations on our dataset. The results demonstrate that our method significantly outperforms these SOTA methods by substantial margins.

## 5.5 ABLATION STUDY

**How does the pruning rate of augmented models affect the attack success rates?**
We experiment with different pruning rate ranges $(r_{min}, r_{max})$ to generate augmented surrogate models. The performance of targeted attacks using ResNet-50 as the original surrogate model and DI-FGSM as the baseline method is presented in Figure 3. The results suggest that excessively high pruning rates can reduce the diversity of intermediate feature maps in surrogate models, which in turn diminishes the transferability of adversarial attacks.

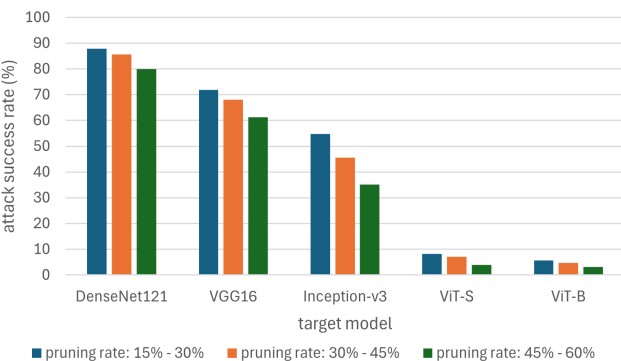

Figure 3: The targeted attacks success rate with different pruning rates of augmented models.

**How do different transformations used in the initial parameter optimization affect the attack success rates?**
We test three types of transformations—rotation, simple adversarial attack, and random masking—for generating augmented models. The performance of untargeted attacks using ResNet-50 as the surrogate model and VNI-FGSM as the baseline method is shown in Figure 4. In simpler cases (e.g., ResNet-50 ⇒ ResNet-101, Inception-v3, Inception-v4), the choice of transformation has minimal impact on transferability. However, transformations that are more challenging for contrastive prediction, such as simple adversarial attacks and random masking, tend to perform better

in more difficult scenarios (e.g., ResNet-50 $\Rightarrow$ ViT-B). This may be because these transformations prompt the pruned models to generate more diverse and informative feature maps.

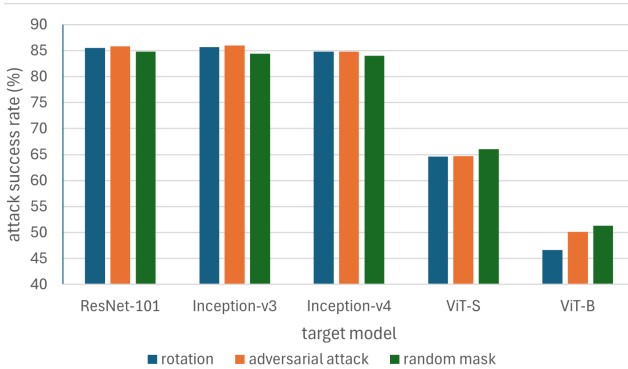

Figure 4: The untargeted attack success rates with different transformations in initial parameter optimization. Harder transformations like simple adversarial attacks and random masking perform better in more difficult scenarios (ResNet-50 $\Rightarrow$ ViT-B).

**How does regularization on different layers affect the attack success rates?**
We apply gradient regularization to the feature maps at different layers. The performance of targeted attacks using ResNet-50 as the surrogate model and DI-FGSM as the baseline method is shown in Figure 5. The results do not show significant differences across layers. However, regularizing the shallower layers tends to yield slightly better performance.

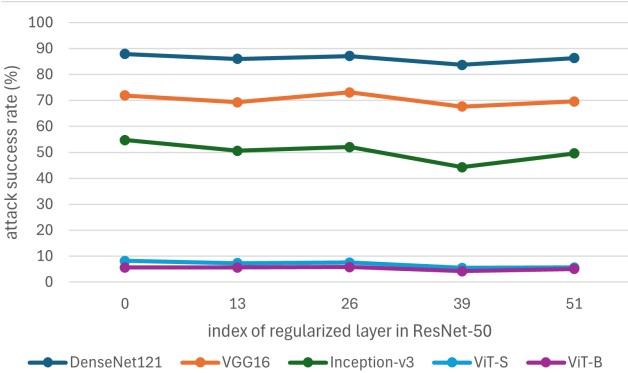

Figure 5: The targeted attacks success rate with regularization on different layers. Regularizing the shallower layers leads to better performance.

## 6    CONCLUSION

In this work, we examine the mechanism of adversarial attacks through the lens of the feature "distribution" across channels. We observe that existing adversarial attacks tend to overly focus on features in a limited number of channels, which hinders their transferability. To address this, we propose a novel model augmentation method that can be integrated with existing attacks. Extensive experiments demonstrate that our approach significantly enhances current attacks and outperforms other model augmentation techniques. We hope our method and perspective provide valuable insights into this complex field and inspire new interpretations of adversarial attack transferability.

## 7    REPRODUCIBILITY

We have submitted the zip file of our source code in the supplementary material.

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
