# OpenReview forum: "Enhance the Transferability of Adversarial Attacks through Channel Pruning"
_ICLR.cc/2025/Conference — ICLR 2025 Conference Withdrawn Submission_

### Official Review · Reviewer_g6Th · 2024-11-01

**Soundness:** 1
**Presentation:** 2
**Contribution:** 2
**Rating:** 3
**Confidence:** 4

**Summary:**

Adversarial examples have recently received much attention in the black-box transfer-based attack scenario due to its more realistic attack setting. To enhance the transferability of the generated adversarial example, the paper introduces GRASP, a model augmentation method that uses channel pruning to generate different models. These different pruned models are used to generate an adversarial example that may be less specific to the potential channel redundancy of the source model.

**Strengths:**

-	Studying the transferability of adversarial examples is an important topic.
-	The paper is easy to follow.

**Weaknesses:**

-	**Experiments are insufficient and do not sufficiently support claims.** The comparison of the proposed method against other model augmentation methods is only done for one source model (ResNet50). Despite the source network being the same for Table 3 and Table 4, the targeted networks are different, which is surprising. The superiority of the proposed method cannot be established. Moreover, since the proposed method uses three pruned networks to generate the adversarial example, I expect the authors to compare their method against an ensemble method to ensure that the gain obtained is not due to ensembling networks.  It would be interesting to compare against ensemble-based methods such as [A] and [B].


- **Motivation of the method.** In the introduction, the authors say, “On the contrary, some
of them are vague and seem to only contain weak features about the object in the input image. If
we only conduct adversarial attacks based on original CNN models, the adversarial samples tend to “overfit” on those highly repetitive features.” I do not understand why, if a feature is not useful for predicting an object (whereas this feature may be useful for another object), this feature would be exploited by an attack that tries to fool the network. I would expect the attack to disrupt features responsible for predicting the target class and not those that are not useful. To validate their intuitions, can the authors provide an experiment or proof showing that adversarial examples tend to “overfit” those highly repetitive, unuseful features? Moreover, I find it strange that increasing the pruning rate, which is the core of the method, degrades the transferability of the generated adversarial examples. Can the authors clarify this point, please?

Typos:
- In Table 3 and 4, there is a blank row.
- Table 1 and 2 are overextended.

[A] Tang, B., Wang, Z., Bin, Y., Dou, Q., Yang, Y., & Shen, H. T. (2024). Ensemble Diversity Facilitates Adversarial Transferability. In Proceedings of the IEEE/CVF Conference on Computer Vision and Pattern Recognition (pp. 24377-24386).

[B] Chen, B., Yin, J., Chen, S., Chen, B., & Liu, X. (2023). An adaptive model ensemble adversarial attack for boosting adversarial transferability. In Proceedings of the IEEE/CVF International Conference on Computer Vision (pp. 4489-4498).

**Questions:**

-	Why did you do not discuss in the related work the loss-based transfer methods such as for example [C] or [D]?
-	For the contrastive learning, how did you choose these three transformations? Why not other ?

[C] Naseer, M., Khan, S., Hayat, M., Khan, F. S., & Porikli, F. (2021). On generating transferable targeted perturbations. In Proceedings of the IEEE/CVF International Conference on Computer Vision (pp. 7708-7717).

[D] Zhao, A., Chu, T., Liu, Y., Li, W., Li, J., & Duan, L. (2023). Minimizing maximum model discrepancy for transferable black-box targeted attacks. In Proceedings of the IEEE/CVF conference on computer vision and pattern recognition (pp. 8153-8162).

---

### Official Review · Reviewer_msvL · 2024-11-02

**Soundness:** 1
**Presentation:** 1
**Contribution:** 1
**Rating:** 3
**Confidence:** 5

**Summary:**

The paper presents a method for a transfer-based blackbox attack.The main idea for the method is to augment the surrogate models by differently pruned models, and generate the adversarial examples. First to augment the models, the model is pruned with the predetermined channel pruning rate. To train the augmented models, the knowledge distillation is used to match the accuracy of the pruned model, incorporating self-supervision and input augmentation as terms in the loss function. The authors present explanations and experiments to further demonstrate their idea. The experiment is done on the ImageNet-like dataset on targeted and untargeted attacks.

**Strengths:**

The paper is easy to read through.

**Weaknesses:**

The weakness of the paper is three-folds.
- The terminology used in the paper lacks objective justification and includes many subjective and inaccurate expressions.
-- See questions.
- Lack of Novelty.
-- The main methodology comes from the idea of channel-pruning based model augmentation.  There are plenty of existing studies that analyze channel-wise pruning in terms of robustness, and numerous studies demonstrate that model augmentation can achieve higher transferability. [1-3]  As it stands, there doesn't seem to be much take-away for the reader from the combined ideas in this study.
- The overall explanation and experiments are insufficient.
-- Lack of Validation to the proposed method.
The experiments are limited to a single dataset, which does not demonstrate generalized results..
he paper primarily explains the performance of the proposed method through intuition and explanation, but lacks supporting evidence. Adding intermediate empirical or theoretical validations for the proposed method would improve its persuasiveness.
To enhance the persuasiveness of this study, more ablation studies are needed.

-[1] Bai et al., "Improving Adversarial Robustness via Channel-wise Activation Suppressing", ICLR 2021
-[2] Borkar et al.,  Defending Against Universal Attacks Through Selective Feature Regeneration, CVPR 2020
-[3] Tramer et al., "Ensemble adversarial training: Attacks and defenses", ICLR 2018

**Questions:**

- Why are some channels redundant? Why can this explained with over-parameterization? (Line 47.)
- Why are the pruned surrogate models containing denser information if initial parameter optimization and regularization applied? (Figure 1)
- Why do samples overfit on the highly repetitive channels?
- How are the hyperparameters set in 4.3?

---

### Official Review · Reviewer_DPG4 · 2024-11-03

**Soundness:** 2
**Presentation:** 2
**Contribution:** 2
**Rating:** 3
**Confidence:** 5

**Summary:**

In this paper, the authors propose a model augmentation-based method to improve the transferability of adversarial attacks. To enhance performance in black-box settings, they first introduce the technique of channel pruning to create a self-ensemble surrogate model, which mitigates the overfitting on redundant features or channels. Additionally, the authors integrate both the knowledge distillation and gradient regularization into this ensemble model to further enhance the transferability across various target models. Experiments conducted on multiple CNN and ViT networks using the ImageNet benchmark dataset demonstrate that the proposed model augmentation method achieves relatively high attack performance.

**Strengths:**

1. The authors propose using channel pruning to enhance the transferability of adversarial attacks, along with knowledge distillation to recover the classification accuracy of pruned models.

2. During the perturbation training, regularization of important feature maps is introduced to reduce the gradient variance to further enhance the attack performance.

3. Experimental results on various target models demonstrate that the proposed transfer-based method outperforms baseline methods in attack effectiveness.

**Weaknesses:**

1. The motivation for introducing channel pruning is not clearly explained. For example, why was channel pruning selected over other pruning techniques, such as kernel or block pruning? Furthermore, as noted in lines 255-257, the L2-norm distance metric lacks robustness due to its sensitivity to outlier noise. Does this criterion maintain consistent performance across different surrogate models?

2. The submitted manuscript lacks certain details regarding initial parameter optimization. For example, the network structure of the self-supervision (SS) module is not specified. Additionally, the authors introduce the  L_{ss}  loss, as defined in lines 295-296, to maximize the similarity of all positive and negative data pairs, which is inconsistent with standard contrastive learning practices.

3. In Section 4.3, the authors propose regularizing gradient variance within an intermediate feature map but do not specify how this layer is selected within the surrogate model. Additionally, as mentioned in lines 322-323, the overall loss function is designed to minimize each loss term including the L_{ce} loss, which may conflict with the goal of crafting adversarial examples by maximizing L_{ce}. Furthermore, the details regarding the optimization of this overall loss function are not clearly explained.

4. As shown in Tables 3-4, the experiment is conducted on only one surrogate model, which does not provide a comprehensive evaluation. Furthermore, the proposed GRASP method is not directly comparable to other model augmentation methods, as these methods do not incorporate knowledge distillation or gradient regularization.

5. The authors claim that reducing gradient variance can balance the importance of different channels within a layer. However, as shown in Figure 5, the minimal change in attack success rate after regularizing different layers does not provide strong support for this claim.

**Questions:**

1. What are the key theoretical differences between pruning techniques like channel, kernel, and block pruning? Do these techniques exhibit different attack performances in both white-box and black-box settings?

2. Could other model augmentation-based attack methods be further improved by incorporating the proposed initial parameter optimization and gradient regularization strategies?

3. Could the authors provide additional details on how adversarial examples are trained on the ensemble of pruned models? In the overall loss function, does knowledge distillation by minimizing L_{ce} conflict with the perturbation training, which involves maximizing L_{ce}? Additionally, how can the proposed five loss terms be effectively trained, and how should their corresponding hyperparameters be adjusted?

4. Could the authors provide additional details on why gradient regularization results in only slight changes in ASR across different layers of the surrogate model? Additionally, why is gradient regularization applied to only one layer rather than multiple layers?

---

### Official Review · Reviewer_CDqA · 2024-11-04

**Soundness:** 2
**Presentation:** 1
**Contribution:** 3
**Rating:** 3
**Confidence:** 5

**Summary:**

This paper proposes a transferable black-box attack using the concept of model augmentation through channel pruning and knowledge distillation. The authors show that the transferability of existing black-box attacks is limited due to their uneven focus on the channels. The authors also introduce a gradient regularization to enhance the transferability further. The evaluation is done using a subset of 2000 images from ImageNet.

**Strengths:**

- The attack is transferable across multiple network architectures, within CNNs as well as from CNNs to transformers.
- Intuitive, novel method for increasing transferability of the black-box attacks.

**Weaknesses:**

- The paper does not follow the format guidelines closely. Tables 1 and 2 are out of the paper margin significantly.
- The presentation of the paper needs to be improved. The paper contains many grammatical errors, like a period in the middle of the sentences and incorrectly capitalized words. For example, on line 354, it should be "Table" instead of "table"; ",," on line 355; should be "pruning" instead of "running" on line 252; the sentence on lines 226-227 is incomplete; there is a blank line in Table 3 and 4 before the first result row.
- What is the size of the test set (out of 2000 images subset) used for evaluation?
- On line 329, the authors mention "almost correctly classified by all the evaluated models" about the chosen subset. What are the exact accuracy numbers for every network?
- Limitations like an increase in computing due to model augmentation and the trade-off between the transferability and number of models should be discussed.

**Questions:**

- What is the size of the test set (out of 2000 images subset) used for evaluation?
- On line 329, the authors mention "almost correctly classified by all the evaluated models" about the chosen subset. What are the exact accuracy numbers for every network?
Refer to weakness section for more comments.

---

### Note · Authors · 2024-11-19

**Comment:**

Thank you for your efforts in providing helpful reviews and suggestions. We will withdraw this submission and keep refining our work.

**Withdrawal Confirmation:**

I have read and agree with the venue's withdrawal policy on behalf of myself and my co-authors.